# *Pneumocystis jirovecii* Pneumonia in Patients with Solid Malignancies: A Retrospective Study in Two Hospitals

**DOI:** 10.3390/pathogens11101169

**Published:** 2022-10-11

**Authors:** Cheon-Hoo Jeon, Si-Ho Kim, Seulki Kim, Moonsuk Bae, Su-Jin Lee, Seungjin Lim

**Affiliations:** 1Division of Infectious Diseases, Department of Internal Medicine, Samsung Changwon Hospital, Sungkyunkwan University School of Medicine, 158, Paryong-ro, Masanhoewon-gu, Changwon 51353, Korea; 2Division of Infectious Disease, Department of Internal Medicine, Pusan National University Yangsan Hospital, 20, Geumo-ro, Mulgeum-eup, Yangsan 50612, Korea; 3Research Institute for Convergence of Biomedical Science and Technology, Pusan National University Yangsan Hospital, 20, Geumo-ro, Mulgeum-eup, Yangsan 50612, Korea

**Keywords:** *Pneumocystis jirovecii* pneumonia, solid malignancies, 30-day survival

## Abstract

*Pneumocystis jirovecii* pneumonia (PJP) is a rare opportunistic infection in patients with solid malignancies. This study aimed to examine the characteristics of patients with solid cancers and PJP. We retrospectively reviewed the medical records of patients with solid tumors and PJP over an 11-year period, enrolling a total of 47 patients (30-day survival group: n = 20, 30-day mortality group: n = 27). Only 34% of patients received ≥20 mg of prednisolone for ≥2 weeks, and the 30-day mortality rate was 57.4%. The 30-day survival group included more women and patients with colon cancer than the mortality group. Furthermore, absolute lymphocyte counts (ALCs) were decreased at PJP symptom onset, as compared with the values observed 1–3 months earlier. Increased oxygen demand and low ALCs after 5–7 days of PJP treatment were also related to poor prognosis. Due to the limitations of this retrospective study, further studies that adhere to the PJP criteria of the European Organization for Research and Treatment of Cancer and the Mycoses Study Group Education and Research Consortium will be needed to evaluate PJP in solid malignancies more clearly.

## 1. Introduction

*Pneumocystis jirovecii* pneumonia (PJP) is an opportunistic infection caused by a fungus [1]. PJP is common in patients with non-human immunodeficiency virus (HIV) who receive immunosuppressive therapy [1]. PJP may develop in patients receiving intensive chemotherapies and immunotherapies, organ transplantation, or corticosteroid treatment [2].

In adult patients with solid malignancies, PJP prophylaxis (trimethoprim/sulfamethoxazole; TMP/SMX) is recommended in several scenarios: (1) after treatment with alemtuzumab for a minimum of 2 months and before CD4+ T-lymphocyte count is ≥200 cells/μL; (2) during active treatment with phosphatidylinositol-3-kinase (PI3K) inhibitors (copanlisib and idelalisib) and/or rituximab; (3) during active treatment with prednisolone (PSL) equivalent to ≥20 mg daily for ≥4 weeks; (4) during temozolomide administration with radiation therapy until recovery from lymphocytopenia; and (5) after purine analog therapy and other T-cell-depleting interventions until CD4 count is ≥200 cells/μL [2]. However, some cancer treatments are risk factors for PJP [3], including radiation therapy, intermittent courses of corticosteroids during chemotherapy, and some chemotherapeutic agents [3]. 

Given the variety of clinical situations and the low incidence of PJP in patients with solid tumors, few multicenter studies enrolled more than 30 patients with solid tumors and PJP [1,4,5,6,7,8]. PJP is difficult to predict. Decreased lymphocyte counts may be associated with PJP in patients with solid malignancies in previous studies [1,9].

In patients with HIV and CD4 T-lymphocyte counts of <200 cells/mm^3^, PJP prophylaxis is recommended. However, CD4 T-lymphocyte count is not routinely evaluated in patients with solid malignancies, thus reducing opportunities for PJP prophylaxis. On the other hand, the absolute lymphocyte counts (ALCs) are often evaluated in such cases. Because it is unknown whether ALCs can help predict PJP onset, this study aimed to examine the characteristics of patients with solid cancers and PJP to identify the risk factors for 30-day mortality in this patient group and evaluate ALCs before and after PJP onset.

## 2. Results

During the 11-year study period, 93,200 patients with solid tumors visited the two study centers. Respiratory specimens of 60 patients were positive for *P. jirovecii* in real-time or conventional polymerase chain reaction (PCR) analysis. A total of 47 patients met the study inclusion criteria, and 13 patients were excluded. The incidence of PJP among patients with solid tumors was 52.57 per 100,000 patients (0.05%).

### 2.1. Baseline Characteristics of PJP Patients

The characteristics of patients with solid tumors before PJP onset are shown in Table 1. The 30-day mortality group included more men than women (81.5% versus 50%, *p =* 0.024). The 30-day survival group included more patients with colon cancer than the 30-day mortality group (*p* = 0.027). All patients with gastric cancer and PJP (n = 5) were in the 30-day mortality group. However, it was not statistically significant between the two groups (*p* = 0.063). Only 34% of patients who received PSL 20 mg or another steroid with an equivalent dose or higher for ≥2 weeks (PJP risk factor) were identified. A total of 44.7% of patients received radiation therapy, and 61.7% of patients had metastatic cancer. These were not significantly different between the two groups. During the study period, no patients were diagnosed with PJP after receiving chemotherapy with alemtuzumab, rituximab, PI3K inhibitor, or temozolomide.

### 2.2. PJP Clinical Characteristics and Laboratory Findings 

Table 2 and Table 3 show the clinical characteristics of patients after the onset of PJP, including laboratory findings, on days 5–7 after treatment initiation. Both groups shared similar symptoms. However, the increase in oxygen demand after compared with that before PJP onset was observed in the 30-day mortality group (*p* = 0.011). In addition, absolute neutrophil counts (ANCs) increased in the 30-day mortality group at the time of PJP onset (*p* = 0.019). Other laboratory values were comparable in both groups, except blood urea nitrogen (BUN) (*p* = 0.011). On days 5–7, laboratory findings revealed that ALCs were higher in the 30-day survival group than in the 30-day mortality group (*p =* 0.003). 

### 2.3. Treatment Courses of PJP

The patients’ clinical characteristics during PJP treatment are shown in Table 4. The time from PJP onset to treatment was comparable in both groups (*p =* 0.878). First-line treatment for PJP was TMP/SMX for most patients (n = 45 (95.7%)). Treatment duration was longer in the survival group than in the mortality group (*p* < 0.001). Patients who survived beyond 30 days were likely to complete PJP treatment (21 days) [10]. The 30-day survival group was more likely than its counterpart to present with TMP/SMX-related side effects (*p* = 0.042); the risk of adverse effects increased with the duration of treatment. The rates of mechanical ventilation use were comparable in both groups. The cause of death within 30 days was exacerbation of PJP in 19 (70.4%) patients and cancer progression in 4 (14.8%) patients (Table 5). In the 30-day mortality group, the median duration of survival after starting treatment for PJP was 10.0 (range; 0–26 days) days.

### 2.4. ALC Changes before and after PJP Onset

ALC values differed between groups on days 5 and 7 after treatment initiation (*p* = 0.003) (Figure 1). In the overall sample, ALC changes were observed over time (*p =* 0.007). In the subgroup analysis, a statistically significant change in ALCs was observed in the 30-day mortality group (*p =* 0.002).

The Wilcoxon signed-rank test was used in the post hoc analysis (Table 6). In the overall sample, there was a significant difference in ALC values between Periods 1 and 3 (*p* < 0.001). In the 30-day survival group, the Friedman test revealed no significant difference between the overall time periods (Table 7). However, significant differences were observed in ALC values between Periods 1 and 3 (*p* = 0.022) and between Periods 3 and 4 (*p* = 0.027) in the post hoc analysis. In the 30-day mortality group, significant differences were observed in ALC values between Periods 1 and 3 and between Periods 1 and 4 (*p =* 0.022, *p* < 0.001, respectively). These results suggest that ALC decreased in both groups 1–3 months before the onset of PJP symptoms (*p* < 0.001). In addition, ALC increased 5–7 days after treatment initiation in the 30-day survival group when compared with the values observed at the time of PJP onset. However, this phenomenon was not observed in the 30-day mortality group (*p* = 0.265). This result suggests that ALC values observed 5–7 days after PJP treatment initiation may help predict 30-day survival (Figure 1).

## 3. Discussion

This study aimed to examine the characteristics of patients with solid cancers and PJP to identify risk factors for 30-day mortality, and to evaluate ALC values before and after PJP onset. Sex distribution, colon cancer rates, BUN, and ANC values differed between the survival and mortality groups. Male patients with lung (n = 12) and gastric (n = 4) cancers had high mortality rates. Consequently, any sex-based prognoses should be interpreted with caution. The increase in oxygen demand after PJP onset was associated with the 30-day mortality rate, suggesting that oxygen demand monitoring may be required during treatment for PJP. 

Notably, this study is important in that the changes were analyzed by examining the patients’ ALCs before PJP symptom onset. ALC declined over time in all patients until onset of PJP. However, significant difference in ALC was observed between two groups at 5–7 days after initiating treatment. In a previous study, CD+ T-cell counts in immunosuppressed persons with or without HIV infection could be indicative of high PCP risk [2,11]. This study finding could be additional evidence that CD4+ T-cell and total lymphocyte count monitoring may be required in patients with solid tumors [3,12].

We calculated the daily steroid dose equivalent as prednisolone (PSL) per patient for 3 months before PJP onset to evaluate steroid administration. Only in 34% of patients who received PSL 20 mg or another steroid with an equivalent dose or higher for ≥ 2 weeks, is this rate comparable to that previously reported (30%) [3]. Therefore, the steroid dosing method was not the only risk factor for PJP in patients with solid tumors.

This study had several limitations. First, the sample size was small, precluding the analysis of cancer type or drug impact on PJP outcomes. Second, this was a retrospective study; thus, T-lymphocyte subset counts and beta D-glucan values could not be evaluated. Third, conventional or real-time PJP PCR analysis was associated with the risk of false positive results. Fourth, due to the limitations of the retrospective study design and the different test method implemented at the study centers, the patients did not fully meet the criteria of probable invasive fungal infection (pneumocystosis), according to the European Organization for Research and Treatment of Cancer and the Mycoses Study Group (EORTC)/Mycoses Study Group Education and Research Consortium (MSGERC) [13]. Fifth, this study lacked a control group of patients with solid tumors and without PJP. A case-control study is warranted. 

These limitations notwithstanding, this study had several strengths. First, this multicenter study suggested the clinical relevance of PJP in solid malignancies, which had high mortality rates. Second, a relatively small proportion of PJP patients received moderate to high doses of steroids. Third, this study showed that a significant ALC reduction before PJP onset and persistent decrease of ALC in the mortality group during PJP treatment. This study also suggests that not only follow-up ALC measurements but also detailed CD4+ T-lymphocyte count measurements are necessary for patients with solid tumors to follow the revised EORTC/MSGERC criteria. Clinicians should be aware that PJP may develop even if the dosage of steroid and duration of administration do not meet the criteria of PJP prophylaxis guidelines. A case-control study is required to evaluate the effect of changes in T-lymphocyte total and subset counts in patients with cancers and PJP.

## 4. Materials and Methods

### 4.1. Study Design and Settings

We retrospectively reviewed the medical records of patients diagnosed with PJP at two tertiary hospitals from 1 January 2010, to 31 December 2020. Patients with solid cancers who visited the study centers were eligible for this study if they met the following all inclusion criteria: (1) age ≧18 years, diagnosis of solid cancer with newly developed or worsening respiratory symptoms (cough, dyspnea, sputum, tachypnea (>20/min), increase in oxygen demand); (2) evidence of new-onset or progressing bilateral pneumonia on chest X-ray or computed tomography; (3) positive PJP PCR or immunofluorescence (Giemsa or methenamine silver stain) *P. jirovecii* test results.

The patients were excluded from this study if they met any of the following criteria: diagnosis of hematological malignancies (lymphoid disorder or myeloid disorder, including lymphoma, leukemia, and multiple myeloma), no radiologic or clinical evidence of pneumonia, or diagnosis of another type of pneumonia or improvement without PJP treatment. The SEE AMP Conventional PCR for *Pneumocystis jirovecii* (Seegene, Seoul, Republic of Korea) was performed at Yangsan Pusan University Hospital. The AmpliSens *Pneumocystis jirovecii (carinii)*-FRT PCR kit (InterLabService Ltd., Moscow, Russia), a real-time PJP PCR test, was used at Samsung Changwon Hospital. The cyclic threshold (Ct) value of the kit was 37.

### 4.2. Study Outcomes 

The primary outcome was the 30-day survival rate after PJP diagnosis. Factors associated with 30-day survival/mortality rates were examined. Causes of death were evaluated, as relevant. 

### 4.3. Statistical Analysis 

Patient characteristics were examined in descriptive analyses. The Mann–Whitney U test was used to compare continuous variables between the survival and mortality patients. The Fisher exact test was used to compare categorical variables. ALC values were examined before and after PJP onset. In Table 6, Periods 1–4 were defined as follows: (1) Period 1, 1–3 months before PJP onset; (2) Period 2, 1–2 weeks before PJP onset; (3) Period 3, time from PJP onset to diagnosis; and (4) Period 4, 5–7 days after PJP initiation. The Friedman test was used to analyze ALC changes over time for the overall sample and for each survival group. The Wilcoxon signed-rank test was used for post hoc analysis. All data were analyzed using IBM SPSS Statistics for Windows version 27.0 (IBM Corp, Armonk, NY, USA). Statistical significance was set at *p*-values of <0.05.

## 5. Conclusions

This study showed that PJP in patients with solid tumors is associated with high mortality rates. In addition, patients who did not receive 20 mg of PSL for ≥2 weeks could develop PJP. Lymphocyte count decline in patients with solid tumors may be a risk factor for PJP, while increased oxygen demand in patients with PJP and persistence of decreased ALC after treatment of PJP may be indicative of poor prognosis. 

## Figures and Tables

**Figure 1 pathogens-11-01169-f001:**
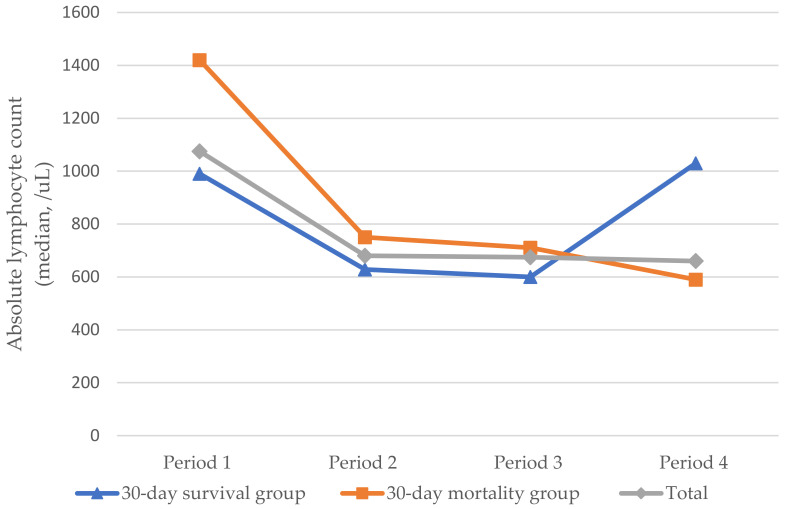
Changes in median absolute lymphocyte count over time in patients with solid tumor and *Pneumocystis jirovecii* pneumonia (Period 1: 1–3 months before PJP onset; Period 2: 1–2 weeks before PJP onset, Period 3: Time from PJP onset to PJP treatment; Period 4: 5–7 days after PJP treatment).

**Table 1 pathogens-11-01169-t001:** Characteristics of solid cancer patients with *Pneumocystis jirovecii* pneumonia (PJP).

	30-Day Survival Group (n = 20)	30-Day Mortality Group (n = 27)	Total (N = 47)	*p*-Value
Age, years (median (range))	65.5 (46–81)	71 (52–86)	69 (46–86)	0.785
Sex (male; n, (%))	10 (50)	22 (81.5)	32 (68)	0.024
Comorbidity (n, (%))				
Hypertension	8 (40.0)	9 (33.3)	17 (36.2)	0.870
Diabetes	6 (30.0)	8 (29.6)	14 (29.8)	0.978
Chronic hepatitis	2 (10.0)	3 (11.1)	5 (10.6)	0.903
Chronic kidney disease	2 (10.0)	3 (11.1)	5 (10.6)	1.000
Type of solid tumor (n, (%))				
Lung cancer	3 (15.0)	11 (40.7)	14 (29.8)	0.056
Breast cancer	4 (20)	2 (7.4)	6 (12.8)	0.201
Gastric cancer	0	5 (18.5)	5 (10.6)	0.063
Colon cancer	4 (20)	0 (0)	4 (8.5)	0.027
Periampullary cancer	1 (5.0)	3 (11.1)	4 (8.5)	0.626
Esophageal cancer	2 (2.0)	1 (3.7)	3 (6.4)	0.567
Prostate cancer	1 (5.0)	2 (7.4)	3 (6.4)	1.000
Mesothelioma	0	2 (7.4)	2 (4.3)	0.500
Renal cell carcinoma	2 (100)	0 (0)	2 (4.3)	0.176
Bladder cancer	0	1 (3.7)	1(2.1)	1.000
Hepatocellular carcinoma	1 (5.0)	0	1 (2.1)	0.426
Tonsillar cancer	1 (5.0)	0	1 (2.1)	0.426
Thyroid cancer	1 (5)	0	1 (2.1)	0.426
ECOG* score of 3 or more (n (%))	2 (10.0)	6 (22.8)	8 (17.0)	0.242
Presence of metastasis (n (%))	13 (65)	16 (59.3)	29 (61.7)	0.463
Total chemotherapy duration (days) (median (range))	51 (0–1335)	62 (0–370)	56 (0–1335)	0.871
Chemotherapy within 3 months before pjp (n (%))	9 (45)	18 (66.7)	27 (57.4)	0.871
Radiation therapy (n (%))	7 (14.9)	14 (51.9)	21 (44.7)	0.251
Prednisolone 20 mg or more daily for ≥2 weeks within 3 months (n (%))	5 (25.0)	11 (40.7)	16 (34)	0.260
Pjp prophylaxis (n (%))	1 (5)	1(2.2)	2 (4.3)	0.849

Abbreviation: ECOG*, Eastern Cooperative Oncology Group.

**Table 2 pathogens-11-01169-t002:** Clinical manifestation and respiratory specimens of the patients.

	30-Day Survival Group (n = 20)	30-Day Mortality Group (n = 27)	Total (N = 47)	*p*-Value
Symptoms (n (%))				
Dyspnea	16 (80.0)	25 (92.6)	41 (87.2)	0.379
Cough	9 (45.0)	14 (51.9)	23 (48.9)	0.643
Sputum	7 (35.0)	15 (55.6)	22 (46.8)	0.238
Fever	14 (70.0)	17 (63.0)	31 (66.0)	0.758
Increase in oxygen demand (n (%))	12 (60.0)	25 (92.6)	37 (78.7)	0.011
Types of respiratory specimen (n (%))				
Induced sputum	12 (60.0)	21 (77.8)	33 (70.2)	0.280
Bronchial washing	1 (5.0)	0 (0.0)	1 (2.1)	0.426
Bronchoalveolar lavage	7 (35.0)	6 (22.2)	13 (27.7)	0.511
Positive PJP PCR* (n (%))	19 (95.0)	27 (100.0)	46 (97.9)	0.426

Abbreviation: PJP, *Pneumocystis jirovecii* pneumonia; PCR*, polymerase chain reaction.

**Table 3 pathogens-11-01169-t003:** Blood test findings of patients in the survival and mortality groups.

	30-Day Survival Group (n = 20)	30-Day Mortality Group (n = 27)	Total (N = 47)	*p*-Value
**Initial Laboratory Test at PJP Onset; Median (range)**				
WBCs * (/µL)	6990 (1900–15,030)	9800 (2030–27,240)	9180 (1900–27,240)	0.084
ANCs ^†^ (/µL)	5530 (1460–12,410)	8730 (1190–25,000)	8030 (1190–25,000)	0.019
ALCs ^‡^ (/µL)	635 (172–2016)	710 (51–1884)	680 (51–2016)	0.561
Hemoglobin (g/dL)	9.5 (6.0–13.6)	9.9 (7.5–14.3)	9.7 (6.0–14.3)	0.170
Platelet (/µL)	216 (21–683)	164 (24–627)	186 (21–683)	0.449
AST ^§^ (IU/L)	37 (16–137)	33 (15–235)	35 (15–235)	0.685
ALT ^∥^ (IU/L)	17.5 (8–201)	24 (8–77)	20 (8–101)	0.563
LDH ^¶^ (U/L)	635 (212–2100)	623 (337–1670)	628.5 (212–2100)	0.979
BUN ** (mg/dL)	13 (7–27)	20 (9–59)	17.15 (7–59)	0.011
Creatinine (mg/dL)	0.74 (0.26–2.36)	0.66 (0.30–1.64)	0.67 (0.26–2.36)	0.907
CRP ^††^ (mg/dL)	11.10 (2.11–24.91)	9.53 (0.75–35.64)	9.81 (0.75–35.64)	0.732
**Follow-up Laboratory Test on Days 5–7 after PJP Treatment; Median (range)**				
WBCs (/µL)	13,710 (4050–36,060)	13,615 (3070–30,420)	13,710 (3070–36,060)	0.826
ANCs (/µL)	11,145 (3230–33,540)	12,545 (490–27,380)	11,925 (490–33,540)	0.599
ALCs (/µL)	1155 (120–3860)	580 (0–1540)	735 (0–3860)	0.003
Hemoglobin (g/dL)	10.6 (7.0–12.8)	10.15 (7.1–16.0)	10.35 (7.0–16.0)	0.784
Platelet (/µL)	246 (29–663)	143.5(36–464)	162 (29–663)	0.994
AST (IU/L)	34.5 (15–141)	27 (15–585)	32 (15–585)	0.060
ALT (IU/L)	21.5 (6–217)	21.0 (7–688)	21 (6–688)	0.651
LDH (U/L)	579 (251–1519)	780 (375–1784)	588 (251–1784)	0.479
BUN (mg/dL)	16.4 (6.6–154)	26.5 (10.0–41.5)	21.6 (6.6–154.0)	0.897
Creatinine (mg/dL)	0.72 (0.25–2.96)	0.80 (0.29–1.62)	0.74 (0.25–2.96)	0.887
CRP (mg/dL)	3.74 (0.26–14.76)	5.05 (0.9–21.83)	5.0 (0.26–21.83)	0.171

Abbreviations: PJP, *Pneumocystis jirovecii* pneumonia; WBCs *, white blood cells; ANCs ^†^, absolute neutrophil counts; ALCs ^‡^, absolute lymphocyte counts; AST ^§^, aspartate aminotransferasee; ALT ^∥^, alanine transaminase; LDH ^¶^, lactate dehydrogenase; BUN**, blood urea nitrogen; CRP ^††^, C-reactive protein.

**Table 4 pathogens-11-01169-t004:** Characteristics of PJP treatment in the 30-day survival and mortality groups.

	30-Day Survival Group (n = 20)	30-Day Mortality Group (n = 27)	Total (N = 47)	*p*-Value
PJP Treatment (n, (%))	20 (100)	25 (92.6)	45 (95.7)	0.500
Treatment Duration (n (%))	20 (4–29)	9.5 (1–25)	12.5 (1–29)	<0.001
Period from PJP Onset to Treatment Initiation (days)	7.0 (1–53)	8.0 (0–32)	8.0 (0–53)	0.878
Adjuvant Steroid for PJP Treatment (n (%))	12 (60.0)	20 (80.0)	32 (71.1)	0.141
Coinfection with Bacterial Pathogen (n (%))	5 (25)	3 (11.1)	8 (17.0)	0.258
Adverse Reaction to TMP/SMX* (n (%))	13 (65.0)	9 (33.3)	22 (46.8)	0.042
Change to Second-Line Drug† of PJP (n (%))	4 (20.0)	4 (16.0)	8 (17.8)	1.000
Mechanical Ventilation (n (%))	7 (35.0)	7 (25.9)	14 (29.8)	0.501

Abbreviation: PJP, *Pneumocystis jirovecii* pneumonia; TMP/SMX*, trimethoprim/sulfamethoxazole. Note: Second-Line Drug† regimen; combination of clindamycin and primaquine.

**Table 5 pathogens-11-01169-t005:** Causes of death in the 30-day mortality group.

Cause of Death	Patients (n = 27 (%))
PJP aggravation	19 (70.4)
Cancer progression	4 (14.8)
Other infection	2 (7.4)
Others	2 (7.4)

Abbreviation: PJP, *Pneumocystis jirovecii* pneumonia.

**Table 6 pathogens-11-01169-t006:** Wilcoxon signed-rank test results in the post hoc analysis of the Friedman test.

	30-Day Survival		30-Day Mortality		Total	
Absolute Lymphocyte Count Difference by Time	Z	*p*-Value	Z	*p*-Value	Z	*p*-Value
Period 2–Period 1	−1.099	0.272	−2.243	0.025 *	−2.473	0.013
Period 3–Period 1	−2.296	0.022 *	−3.027	0.002 **	−3.382	<0.001 ***
Period 4–Period 1	−0.402	0.687	−3.750	<0.001 **	−3.119	0.002 **
Period 3–Period 2	−0.724	0.469	−0.955	0.339	−1.257	0.209
Period 4–Period 2	−1.811	0.070	−1.542	0.123	−0.152	0.879
Period 4–Period 3	−2.213	0.027*	−1.114	0.265	−0.978	0.328

* *p* < 0.05, ** *p* < 0.01, *** *p* < 0.001.

**Table 7 pathogens-11-01169-t007:** Friedman test for absolute lymphocyte count change over time in patients with solid tumors and PJP.

		Period 1	Period 2	Period 3	Period 4	*p*-Value
		1–3 Months before PJP	1–2 Weeks before PJP	From PJP Onset to Treatment	5–7 Days after PJP Treatment	
30-Day Survival Group (n = 15)	Median (range)	990.0 (350.0–2370.0)	628.0 (380.0–1630.0)	600.0 (180.0–2016.0)	1030.0 (120.0–3860.0)	0.163
	Average rank	3.10	2.50	2.08	2.32	
30-Day Mortality Group (n = 21)	Median (range)	1420.00 (141.0–3690.0)	750.0 (96.8–2370)	710.0 (51.0–1884.0)	590.0 (0–1540)	0.002 **
	Average rank	3.33	2.57	2.19	1.90	
Total (n = 36)	Median (range)	1075.00 (141.0–3690.0)	679.8 (96.8–2370)	674.50 (51–2016)	660.0 (0–3860)	0.007 **
	Average rank	3.10	2.50	2.08	2.32	

Abbreviation: PJP, *Pneumocystis jirovecii* pneumonia. ** *p* < 0.01.

## Data Availability

The data presented in this study are available on request from the corresponding author.

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
