# Peer review of "Pneumocystis jirovecii* Pneumonia in Patients with Solid Malignancies: A Retrospective Study in Two Hospitals"

_pathogens, 2022, doi:10.3390/pathogens11101169_

Round 1
Reviewer 1 Report
The manuscript analyses Pneumocystis jirovecii pneumonia (PJP) in adult patients with solid malignancies. The number of patients is relatively small(47), and the authors analyzed them by comparing two groups: (1) survival group: n = 20, and (2) mortality group: n = 27, looking at many different variables in search of any differences. Discussion should be improved. The authors should reduce unnecessary analysis and bring more novelty to the table.
Author Response
Author’s Reply to Reviewer 1
Thanks for your thoughtful comments. We revised the manuscript as indicated below.
We rechecked the use of English language and highlighted the changes.
- Introduction
We revised the introduction to express this clearly.
Page 2, lines 50-55
“In patients with HIV and CD4 T-lymphocyte counts of <200 cells/mm3, PJP prophylaxis is recommended. However, CD4 T-lymphocyte count is not routinely evaluated in patients with solid malignancies, thus reducing opportunities for PJP prophylaxis. On the other hand, the absolute lymphocyte counts (ALCs) are often evaluated in such cases. Because it is unknown whether ALCs can help predict PJP onset, this study aimed to examine the characteristics of patients with solid cancers and PJP to identify the risk factors for 30-day mortality in this patient group and evaluate ALCs before and after PJP onset.”
- The authors should reduce unnecessary analysis and bring more novelty to the table.
We revised the tables and omitted items for which statistical significance was not confirmed.
Also, we tried to describe this study more clearly by revising the order of items on the tables.
Table 1 Characteristics of solid cancer patients with Pneumocystis jirovecii pneumonia
|
|
30-day survival group (n = 20) |
30-day mortality group (n = 27) |
Total (N = 47) |
P-value |
|
Age, years [median (range)] |
65.5 (46–81) |
71 (52–86) |
69 (46–86) |
0.785 |
|
Sex [male; n, (%)] |
10 (50) |
22 (81.5) |
32 (68) |
0.024 |
|
Comorbidity [n, (%)] |
|
|
|
|
|
Hypertension |
8 (40.0) |
9 (33.3) |
17 (36.2) |
0.870 |
|
Diabetes |
6 (30.0) |
8 (29.6) |
14 (29.8) |
0.978 |
|
Chronic hepatitis |
2 (10.0) |
3 (11.1) |
5 (10.6) |
0.903 |
|
Chronic kidney disease |
2 (10.0) |
3 (11.1) |
5 (10.6) |
1.000 |
|
Type of solid tumor [n, (%)] |
|
|
|
|
|
Lung cancer |
3 (15.0) |
11 (40.7) |
14 (29.8) |
0.056 |
|
Breast cancer |
4 (20) |
2 (7.4) |
6 (12.8) |
0.201 |
|
Gastric cancer |
0 |
5 (18.5) |
5 (10.6) |
0.063 |
|
Colon cancer |
4 (20) |
0 (0) |
4 (8.5) |
0.027 |
|
Periampullary cancer |
1 (5.0) |
3 (11.1) |
4 (8.5) |
0.626 |
|
Esophageal cancer |
2 (2.0) |
1 (3.7) |
3 (6.4) |
0.567 |
|
Prostate cancer |
1 (5.0) |
2 (7.4) |
3 (6.4) |
1.000 |
|
Mesothelioma |
0 |
2 (7.4) |
2 (4.3) |
0.500 |
|
Renal cell carcinoma |
2 (100) |
0 (0) |
2 (4.3) |
0.176 |
|
Bladder cancer |
0 |
1 (3.7) |
1(2.1) |
1.000 |
|
Hepatocellular carcinoma |
1 (5.0) |
0 |
1 (2.1) |
0.426 |
|
Tonsillar cancer |
1 (5.0) |
0 |
1 (2.1) |
0.426 |
|
Thyroid cancer |
1 (5) |
0 |
1 (2.1) |
0.426 |
|
ECOG* score of 3 or more [n (%)] |
2 (10.0) |
6 (22.8) |
8 (17.0) |
0.242 |
|
Presence of metastasis [n (%)] |
13 (65) |
16 (59.3) |
29 (61.7) |
0.463 |
|
Total chemotherapy duration (days), [median (range)] |
51 (0–1335) |
62 (0–370) |
56 (0–1335) |
0.871 |
|
Chemotherapy within 3 months before PJP [n (%)] |
9 (45) |
18 (66.7) |
27 (57.4) |
0.871 |
|
Radiation therapy [n (%)] |
7 (14.9) |
14 (51.9) |
21 (44.7) |
0.251 |
|
Prednisolone 20 mg or more daily for ≥2 weeks within 3 months [n (%)] |
5 (25.0) |
11 (40.7) |
16 (34) |
0.260 |
|
PJP prophylaxis [n (%)] |
1 (5) |
1(2.2) |
2 (4.3) |
0.849 |
Table 2. Clinical manifestation and respiratory specimens of the patients
|
|
30-day survival group (n = 20) |
30-day mortality group (n = 27) |
Total (N = 47) |
P-value |
|
Symptoms [n (%)] |
|
|
|
|
|
Dyspnea |
16 (80.0) |
25 (92.6) |
41 (87.2) |
0.379 |
|
Cough |
9 (45.0) |
14 (51.9) |
23 (48.9) |
0.643 |
|
Sputum |
7 (35.0) |
15 (55.6) |
22 (46.8) |
0.238 |
|
Fever |
14 (70.0) |
17 (63.0) |
31 (66.0) |
0.758 |
|
Increase in oxygen demand [n (%)] |
12 (60.0) |
25 (92.6) |
37 (78.7) |
0.011 |
|
Types of respiratory specimen [n (%)] |
|
|
|
|
|
Induced sputum |
12 (60.0) |
21 (77.8) |
33 (70.2) |
0.280 |
|
Bronchial washing |
1 (5.0) |
0 (0.0) |
1 (2.1) |
0.426 |
|
Bronchoalveolar lavage |
7 (35.0) |
6 (22.2) |
13 (27.7) |
0.511 |
|
Positive PJP PCR* [n (%)] |
19 (95.0) |
27 (100.0) |
46 (97.9) |
0.426 |
Abbreviation: PJP, Pneumocystis jirovecii pneumonia; PCR*, polymerase chain reaction.
Table 3. Blood test findings of patients in the survival and mortality groups
|
|
30-day survival group (n = 20) |
30-day mortality group (n = 27) |
Total (N = 47) |
P-value |
|
Initial laboratory test at PJP onset; median (range) |
||||
|
WBCs* (/µL) |
6990 (1900–15030) |
9800 (2030–27240) |
9180 (1900–27240) |
0.084 |
|
ANCs† (/µL) |
5530 (1460–12410) |
8730 (1190–25000) |
8030 (1190–25000) |
0.019 |
|
ALCs‡ (/µL) |
635 (172–2016) |
710 (51–1884) |
680 (51–2016) |
0.561 |
|
Hemoglobin (g/dL) |
9.5 (6.0–13.6) |
9.9 (7.5–14.3) |
9.7 (6.0–14.3) |
0.170 |
|
Platelet (/µL) |
216 (21–683) |
164 (24–627) |
186 (21–683) |
0.449 |
|
AST§ (IU/L) |
37 (16–137) |
33 (15–235) |
35 (15–235) |
0.685 |
|
ALT∥ (IU/L) |
17.5 (8–201) |
24 (8–77) |
20 (8–101) |
0.563 |
|
LDH¶ (U/L) |
635 (212–2100) |
623 (337–1670) |
628.5 (212–2100) |
0.979 |
|
BUN** (mg/dL) |
13 (7–27) |
20 (9–59) |
17.15 (7–59) |
0.011 |
|
Creatinine (mg/dL) |
0.74 (0.26–2.36) |
0.66 (0.30–1.64) |
0.67 (0.26–2.36) |
0.907 |
|
CRP†† (mg/dL) |
11.10 (2.11–24.91) |
9.53 (0.75–35.64) |
9.81 (0.75–35.64) |
0.732 |
|
Follow-up laboratory test on days 5–7 after PJP treatment; median (range) |
||||
|
WBCs (/µL) |
13710 (4050–36060) |
13615 (3070–30420) |
13710 (3070–36060) |
0.826 |
|
ANCs (/µL) |
11145 (3230–33540) |
12545 (490–27380) |
11925 (490–33540) |
0.599 |
|
ALCs (/µL) |
1155 (120–3860) |
580 (0–1540) |
735 (0–3860) |
0.003 |
|
Hemoglobin (g/dL) |
10.6 (7.0–12.8) |
10.15 (7.1–16.0) |
10.35 (7.0–16.0) |
0.784 |
|
Platelet (/µL) |
246 (29–663) |
143.5(36–464) |
162 (29–663) |
0.994 |
|
AST (IU/L) |
34.5 (15–141) |
27 (15–585) |
32 (15–585) |
0.060 |
|
ALT (IU/L) |
21.5 (6–217) |
21.0 (7–688) |
21 (6–688) |
0.651 |
|
LDH (U/L) |
579 (251–1519) |
780 (375–1784) |
588 (251–1784) |
0.479 |
|
BUN (mg/dL) |
16.4 (6.6–154) |
26.5 (10.0–41.5) |
21.6 (6.6–154.0) |
0.897 |
|
Creatinine (mg/dL) |
0.72 (0.25–2.96) |
0.80 (0.29–1.62) |
0.74 (0.25–2.96) |
0.887 |
|
CRP (mg/dL) |
3.74 (0.26–14.76) |
5.05 (0.9–21.83) |
5.0 (0.26–21.83) |
0.171 |
Abbreviations: PJP, Pneumocystis jirovecii pneumonia; WBCs*, white blood cells; ANCs†, absolute neutrophil counts; ALCs‡, absolute lymphocyte counts; AST§, aspartate aminotransferasee; ALT∥, alanine transaminase; LDH¶, lactate dehydrogenase; BUN**, blood urea nitrogen; CRP††, C-reactive protein.
Table 4. Characteristics of PJP treatment in the 30-day survival and mortality groups
|
|
30-day survival group (n = 20) |
30-day mortality group (n = 27) |
Total (N = 47) |
P-value |
|
PJP treatment [n, (%)] |
20 (100) |
25 (92.6) |
45 (95.7) |
0.500 |
|
Treatment duration [n (%)] |
20 (4-29) |
9.5 (1-25) |
12.5 (1-29) |
<0.001 |
|
Period from PJP onset to treatment initiation (days) |
7.0 (1-53) |
8.0 (0-32) |
8.0 (0-53) |
0.878 |
|
Adjuvant steroid for PJP treatment [n (%)] |
12 (60.0) |
20 (80.0) |
32 (71.1) |
0.141 |
|
Coinfection with bacterial pathogen [n (%)] |
5 (25) |
3 (11.1) |
8 (17.0) |
0.258 |
|
Adverse reaction to TMP/SMX* [n (%)] |
13 (65.0) |
9 (33.3) |
22 (46.8) |
0.042 |
|
Change to second-line drug† of PJP [n (%)] |
4 (20.0) |
4 (16.0) |
8 (17.8) |
1.000 |
|
Mechanical ventilation [n (%)] |
7 (35.0) |
7 (25.9) |
14 (29.8) |
0.501 |
Abbreviation: PJP, Pneumocystis jirovecii pneumonia; TMP/SMX*, trimethoprim/sulfamethoxazole.
Note: second-line drug regimen; combination of clindamycin and primaquine
Table 5. Causes of death in the 30-day mortality group
|
Cause of death |
Patients (n = 27, %) |
|
PJP aggravation |
19 (70.4) |
|
Cancer progression |
4 (14.8) |
|
Other infection |
2 (7.4) |
|
Others |
2 (7.4) |
Abbreviation: PJP, Pneumocystis jirovecii pneumonia
- Discussion should be improved.
We modified the manuscript briefly to make it clear. It was modified as follow:
Pages 7-9, lines 141-215
“3. Discussion
- Discussion
This study aimed to examine the characteristics of patients with solid cancers and PJP to identify risk factors for 30-day mortality, and to evaluate ALC values before and after PJP onset. Sex distribution, colon cancer rates, BUN, and ANC values differed between the survival and mortality groups. Male patients with lung (n = 12) and gastric (n = 4) cancers had high mortality rates. Consequently, any sex-based prognoses should be interpreted with caution. The increase in oxygen demand after PJP onset was associated with the 30-day mortality rate, suggesting that oxygen demand monitoring may be required during treatment for PJP.
Notably, this study is important in that the changes were analyzed by examining the patients’ ALCs before PJP symptom onset. ALC declined over time in all patients until onset of PJP. However, significant difference in ALC was observed between two groups at 5–7 days after initiating treatment. In a previous study, CD+ T-cell counts in immunosuppressed persons with or without HIV infection could be indicative of high PCP risk [2, 11]. This study finding could be additional evidence that CD4+ T-cell and total lymphocyte count monitoring may be required in patients with solid tumors [3, 12].
We calculated the daily steroid dose equivalent as prednisolone (PSL) per patient for 3 months before PJP onset to evaluate steroid administration. Only in 34% of patients who received PSL 20 mg or another steroid with an equivalent dose or higher for ≥2 weeks, is this rate comparable to that previously reported (30%) [3]. Therefore, the steroid dosing method was not the only risk factor for PJP in patients with solid tumors.
This study had several limitations. First, the sample size was small, precluding the analysis of cancer type or drug impact on PJP outcomes. Second, this was a retrospective study; thus, T-lymphocyte subset counts and beta D-glucan values could not be evaluated. Third, real-time PJP PCR analysis was associated with the risk of false positive results. Fourth, this study lacked a control group of patients with solid tumors and without PJP. A case–control study is warranted.
These limitations notwithstanding, this study had several strengths. First, this multicenter study suggested the clinical relevance of PJP in solid malignancies, which had high mortality rates. Second, a relatively small proportion of PJP patients received moderate to high dose of steroids. Third, this study showed that a significant ALC reduction before PJP onset and persistent decrease of ALC in the mortality group during PJP treatment. Clinicians should be aware that PJP may develop even if the dosage of steroid and duration of administration do not meet the criteria of PJP prophylaxis guidelines. A case–control study is required to evaluate the effect of changes in T-lymphocyte total and subset counts in patients with cancers and PJP.
- Conclusions
This study showed that PJP in patients with solid tumors is associated with high mortality rates. In addition, patients who did not receive 20 mg of PSL for ≥2 weeks could develop PJP. Lymphocyte count decline in patients with solid tumors may be a risk factor for PJP, while increased oxygen demand in patients with PJP and persistence of decreased ALC after treatment of PJP may be indicative of poor prognosis.”

Reviewer 2 Report
Pneumocystis jirovecii infection is characteristic of AIDS patients. However, it is often forgotten that Pneumocystis jirovecii pneumonia will also appear in other immunocompromised patients, including cancer. Therefore, the presented article is medically important. Patients are well characterized. The results show that Pneumocystis jirovecii pneumonia infection occurs in approximately 1 in 2,000 people with solid tumor.
I have one caveat. The description of the PCR method should be in Chapter 4. Materials and Methods, not Results.
Author Response
Author’s Reply to the Reviewer 2
Thanks for your thoughtful comment.
English language and style are fine/minor spell check required
We rechecked the manuscript to ensure appropriate English language usage and style.
I have one caveat. The description of the PCR method should be in Chapter 4. Materials and Methods, not Results.
We moved the sentences to section 4.
Page 8, lines 179-196
- Materials and Methods
4.1. Study design and settings
We retrospectively reviewed medical records of patients diagnosed with PJP at two tertiary hospitals from January 1, 2010, to December 31, 2020. Patients with solid cancers who visited study centers were eligible for this study if they met the following all inclusion criteria: 1) age≧18 years, diagnosis of solid cancer with newly developed or worsening respiratory symptoms (cough, dyspnea, sputum, tachypnea [>20/min], increase in oxygen demand), 2) evidence of new-onset or progressing bilateral pneumonia on chest X-ray or computed tomography, 3) positive PJP PCR or immunofluorescence (Giemsa or methenamine silver stain) P. jirovecii test results.
The patients were excluded from this study if they met any of the following criteria: diagnosis of hematological malignancies (lymphoid disorder or myeloid disorder, including lymphoma, leukemia, and multiple myeloma), no radiologic or clinical evidence of pneumonia, diagnosis of another type of pneumonia or improvement without PJP treatment. Tests for P. jirovecii PCR (SEE AMP; Seegene, Seoul, Republic of Korea) were performed at the Yangsan Pusan University hospital. AmpliSens Pneumocystis jirovecii (carinii)-FRT PCR kit (InterLabService Ltd., Moscow, Russia) was used at the Samsung Changwon Hospital.

Reviewer 3 Report
The manuscript from Jeon, et al. is a retrospective study examining Pneumocystis jirovecii pneumonia (PJP) in patients with solid organ tumors. The goal of the work was to find clinical markers of poor prognosis in hopes of determining when to provide prophylaxis. The authors broke patients up into two groups who had PJP, those that survived and those who succumbed to infection. They had longitudinal data and were able to examine clinical variables prior to PJP diagnosis, at diagnosis, and after diagnosis. Unfortunately, they did not have CD4 T cell counts as these were not done as standard of care. Overall, they found that a drop in absolute lymphocyte counts at 5-7 days after PJP treatment initiation was significantly associated with mortality.
A significant problem with the study, in addition to lack of T cell data, was the lack of data from a control (uninfected) group but more importantly, the small number of patients in each group (20 survival and 27 mortality). The study seems significantly underpowered making it hard to draw conclusions other than a prospective study with a control group and higher n is warranted. Even though the authors indicate the incidence of PJP in solid organ malignancies is higher than they thought, it is still relatively rare. They were only able to find 47 total patients in 10 years using two medical centers. It seems unlikely they would be able to get higher numbers in a reasonable amount of time.
Overall, the manuscript is well written and the weaknesses acknowledged. The tables should include legends explaining what the data represents (mean, median, range, percent, number, etc). For example, in Table 1 it was not clear what the numbers in outside of and inside of the parentheses means.
Author Response
Author’s Reply to Reviewer 3
Thanks for your thoughtful comments. We revised the manuscript as indicated below.
- A significant problem with the study, in addition to lack of T cell data, was the lack of data from a control (uninfected) group but more importantly, the small number of patients in each group (20 survival and 27 mortality). The study seems significantly underpowered making it hard to draw conclusions other than a prospective study with a control group and higher n is warranted. Even though the authors indicate the incidence of PJP in solid organ malignancies is higher than they thought, it is still relatively rare. They were only able to find 47 total patients in 10 years using two medical centers. It seems unlikely they would be able to get higher numbers in a reasonable amount of time.
- We agree with your opinion. The limitations of this study were described in discussion section.
- The tables should include legends explaining what the data represents (mean, median, range, percent, number, etc). For example, in Table 1 it was not clear what the numbers in outside of and inside of the parentheses means.
- We revised the tables as follow.
Table 1 Characteristics of solid cancer patients with Pneumocystis jirovecii pneumonia
|
|
30-day survival group (n = 20) |
30-day mortality group (n = 27) |
Total (N = 47) |
P-value |
|
Age, years [median (range)] |
65.5 (46–81) |
71 (52–86) |
69 (46–86) |
0.785 |
|
Sex [male; n, (%)] |
10 (50) |
22 (81.5) |
32 (68) |
0.024 |
|
Comorbidity [n, (%)] |
|
|
|
|
|
Hypertension |
8 (40.0) |
9 (33.3) |
17 (36.2) |
0.870 |
|
Diabetes |
6 (30.0) |
8 (29.6) |
14 (29.8) |
0.978 |
|
Chronic hepatitis |
2 (10.0) |
3 (11.1) |
5 (10.6) |
0.903 |
|
Chronic kidney disease |
2 (10.0) |
3 (11.1) |
5 (10.6) |
1.000 |
|
Type of solid tumor [n, (%)] |
|
|
|
|
|
Lung cancer |
3 (15.0) |
11 (40.7) |
14 (29.8) |
0.056 |
|
Breast cancer |
4 (20) |
2 (7.4) |
6 (12.8) |
0.201 |
|
Gastric cancer |
0 |
5 (18.5) |
5 (10.6) |
0.063 |
|
Colon cancer |
4 (20) |
0 (0) |
4 (8.5) |
0.027 |
|
Periampullary cancer |
1 (5.0) |
3 (11.1) |
4 (8.5) |
0.626 |
|
Esophageal cancer |
2 (2.0) |
1 (3.7) |
3 (6.4) |
0.567 |
|
Prostate cancer |
1 (5.0) |
2 (7.4) |
3 (6.4) |
1.000 |
|
Mesothelioma |
0 |
2 (7.4) |
2 (4.3) |
0.500 |
|
Renal cell carcinoma |
2 (100) |
0 (0) |
2 (4.3) |
0.176 |
|
Bladder cancer |
0 |
1 (3.7) |
1(2.1) |
1.000 |
|
Hepatocellular carcinoma |
1 (5.0) |
0 |
1 (2.1) |
0.426 |
|
Tonsillar cancer |
1 (5.0) |
0 |
1 (2.1) |
0.426 |
|
Thyroid cancer |
1 (5) |
0 |
1 (2.1) |
0.426 |
|
ECOG* score of 3 or more [n (%)] |
2 (10.0) |
6 (22.8) |
8 (17.0) |
0.242 |
|
Presence of metastasis [n (%)] |
13 (65) |
16 (59.3) |
29 (61.7) |
0.463 |
|
Total chemotherapy duration (days), [median (range)] |
51 (0–1335) |
62 (0–370) |
56 (0–1335) |
0.871 |
|
Chemotherapy within 3 months before PJP [n (%)] |
9 (45) |
18 (66.7) |
27 (57.4) |
0.871 |
|
Radiation therapy [n (%)] |
7 (14.9) |
14 (51.9) |
21 (44.7) |
0.251 |
|
Prednisolone 20 mg or more daily for ≥2 weeks within 3 months [n (%)] |
5 (25.0) |
11 (40.7) |
16 (34) |
0.260 |
|
PJP prophylaxis [n (%)] |
1 (5) |
1(2.2) |
2 (4.3) |
0.849 |
Table 2. Clinical manifestation and respiratory specimens of the patients
|
|
30-day survival group (n = 20) |
30-day mortality group (n = 27) |
Total (N = 47) |
P-value |
|
Symptoms [n (%)] |
|
|
|
|
|
Dyspnea |
16 (80.0) |
25 (92.6) |
41 (87.2) |
0.379 |
|
Cough |
9 (45.0) |
14 (51.9) |
23 (48.9) |
0.643 |
|
Sputum |
7 (35.0) |
15 (55.6) |
22 (46.8) |
0.238 |
|
Fever |
14 (70.0) |
17 (63.0) |
31 (66.0) |
0.758 |
|
Increase in oxygen demand [n (%)] |
12 (60.0) |
25 (92.6) |
37 (78.7) |
0.011 |
|
Types of respiratory specimen [n (%)] |
|
|
|
|
|
Induced sputum |
12 (60.0) |
21 (77.8) |
33 (70.2) |
0.280 |
|
Bronchial washing |
1 (5.0) |
0 (0.0) |
1 (2.1) |
0.426 |
|
Bronchoalveolar lavage |
7 (35.0) |
6 (22.2) |
13 (27.7) |
0.511 |
|
Positive PJP PCR* [n (%)] |
19 (95.0) |
27 (100.0) |
46 (97.9) |
0.426 |
Table 3. Blood test findings of patients in the survival and mortality groups
|
|
30-day survival group (n = 20) |
30-day mortality group (n = 27) |
Total (N = 47) |
P-value |
|
Initial laboratory test at PJP onset; median (range) |
||||
|
WBCs* (/µL) |
6990 (1900–15030) |
9800 (2030–27240) |
9180 (1900–27240) |
0.084 |
|
ANCs† (/µL) |
5530 (1460–12410) |
8730 (1190–25000) |
8030 (1190–25000) |
0.019 |
|
ALCs‡ (/µL) |
635 (172–2016) |
710 (51–1884) |
680 (51–2016) |
0.561 |
|
Hemoglobin (g/dL) |
9.5 (6.0–13.6) |
9.9 (7.5–14.3) |
9.7 (6.0–14.3) |
0.170 |
|
Platelet (/µL) |
216 (21–683) |
164 (24–627) |
186 (21–683) |
0.449 |
|
AST§ (IU/L) |
37 (16–137) |
33 (15–235) |
35 (15–235) |
0.685 |
|
ALT∥ (IU/L) |
17.5 (8–201) |
24 (8–77) |
20 (8–101) |
0.563 |
|
LDH¶ (U/L) |
635 (212–2100) |
623 (337–1670) |
628.5 (212–2100) |
0.979 |
|
BUN** (mg/dL) |
13 (7–27) |
20 (9–59) |
17.15 (7–59) |
0.011 |
|
Creatinine (mg/dL) |
0.74 (0.26–2.36) |
0.66 (0.30–1.64) |
0.67 (0.26–2.36) |
0.907 |
|
CRP†† (mg/dL) |
11.10 (2.11–24.91) |
9.53 (0.75–35.64) |
9.81 (0.75–35.64) |
0.732 |
|
Follow-up laboratory test on days 5–7 after PJP treatment; median (range) |
||||
|
WBCs (/µL) |
13710 (4050–36060) |
13615 (3070–30420) |
13710 (3070–36060) |
0.826 |
|
ANCs (/µL) |
11145 (3230–33540) |
12545 (490–27380) |
11925 (490–33540) |
0.599 |
|
ALCs (/µL) |
1155 (120–3860) |
580 (0–1540) |
735 (0–3860) |
0.003 |
|
Hemoglobin (g/dL) |
10.6 (7.0–12.8) |
10.15 (7.1–16.0) |
10.35 (7.0–16.0) |
0.784 |
|
Platelet (/µL) |
246 (29–663) |
143.5(36–464) |
162 (29–663) |
0.994 |
|
AST (IU/L) |
34.5 (15–141) |
27 (15–585) |
32 (15–585) |
0.060 |
|
ALT (IU/L) |
21.5 (6–217) |
21.0 (7–688) |
21 (6–688) |
0.651 |
|
LDH (U/L) |
579 (251–1519) |
780 (375–1784) |
588 (251–1784) |
0.479 |
|
BUN (mg/dL) |
16.4 (6.6–154) |
26.5 (10.0–41.5) |
21.6 (6.6–154.0) |
0.897 |
|
Creatinine (mg/dL) |
0.72 (0.25–2.96) |
0.80 (0.29–1.62) |
0.74 (0.25–2.96) |
0.887 |
|
CRP (mg/dL) |
3.74 (0.26–14.76) |
5.05 (0.9–21.83) |
5.0 (0.26–21.83) |
0.171 |
Table 4. Characteristics of PJP treatment in the 30-day survival and mortality groups
|
|
30-day survival group (n = 20) |
30-day mortality group (n = 27) |
Total (N = 47) |
P-value |
|
PJP treatment [n, (%)] |
20 (100) |
25 (92.6) |
45 (95.7) |
0.500 |
|
Treatment duration [n (%)] |
20 (4-29) |
9.5 (1-25) |
12.5 (1-29) |
<0.001 |
|
Period from PJP onset to treatment initiation (days) |
7.0 (1-53) |
8.0 (0-32) |
8.0 (0-53) |
0.878 |
|
Adjuvant steroid for PJP treatment [n (%)] |
12 (60.0) |
20 (80.0) |
32 (71.1) |
0.141 |
|
Coinfection with bacterial pathogen [n (%)] |
5 (25) |
3 (11.1) |
8 (17.0) |
0.258 |
|
Adverse reaction to TMP/SMX* [n (%)] |
13 (65.0) |
9 (33.3) |
22 (46.8) |
0.042 |
|
Change to second-line drug† of PJP [n (%)] |
4 (20.0) |
4 (16.0) |
8 (17.8) |
1.000 |
|
Mechanical ventilation [n (%)] |
7 (35.0) |
7 (25.9) |
14 (29.8) |
0.501 |
Table 5. Causes of death in the 30-day mortality group
|
Cause of death |
Patients (n = 27, %) |
|
PJP aggravation |
19 (70.4) |
|
Cancer progression |
4 (14.8) |
|
Other infection |
2 (7.4) |
|
Others |
2 (7.4) |
